# Clay Nanotubes Loaded with Diazepam or Xylazine Permeate the Brain through Intranasal Administration in Mice

**DOI:** 10.3390/ijms24119648

**Published:** 2023-06-02

**Authors:** Yaswanthi Yanamadala, Mahdi Y. Saleh, Afrika A. Williams, Yuri Lvov, Teresa A. Murray

**Affiliations:** 1Center for Biomedical Engineering and Rehabilitation Sciences, Louisiana Tech University, Rustom, LA 71270, USA; yashwanthi789@gmail.com (Y.Y.); afrika.williams@gmail.com (A.A.W.); 2Institute for Micromanufacturing, Louisiana Tech University, Rustom, LA 71270, USA; mahdisaleh95@gmail.com

**Keywords:** halloysite, nanotubes, drug delivery, blood–brain barrier, nanocarriers, intranasal delivery

## Abstract

The blood–brain barrier (BBB) is an obstacle to the permeation of most therapeutic drugs into the brain, limiting treatments for neurological disorders. Drugs loaded within nanocarriers that pass through the BBB can overcome this limitation. Halloysite consists of naturally occurring biocompatible clay nanotubes of 50 nm diameter and 15 nm lumen, allowing the loading and sustained release of loaded drugs. These have demonstrated the ability to transport loaded molecules into cells and organs. We propose to use halloysite nanotubes as a “nano-torpedo” for drug delivery through the BBB due to their needle-like shape. To determine if they can cross the BBB using a non-invasive, clinically translatable route of administration, we loaded halloysite with either diazepam or xylazine and delivered these intranasally to mice daily over six days. The sedative effects of these drugs were observed in vestibulomotor tests conducted at two, five, and seven days after the initial administration. Behavioral tests were conducted 3.5 h after administration to show that the effects were from halloysite/delivered drugs and not from the drug alone. As expected, the treated mice performed more poorly than the sham, drug alone, and halloysite-vehicle-treated mice. These results confirm that halloysite permeates the BBB to deliver drugs when administered intranasally.

## 1. Introduction

Most medications that could treat central nervous system (CNS) disorders are hindered by the physio-chemical features of the blood–brain barrier (BBB). The BBB can entirely prevent the permeability of certain drugs, allow only sub-therapeutic doses to reach the brain or even chemically altering drugs, all of which diminish the therapeutic efficacy of drugs [1]. Biomedical nanotechnology facilitates the creation of novel delivery systems for drugs that would otherwise be ineffective in treating brain disorders [2]. With the increase in longevity and persistence of environmental toxins, a steadily increasing number of people have CNS disorders [3]. This is reflected in the findings from a 2021 global study that the number of people with neurological disorders has surpassed cardiovascular diseases and cancers [4]. This situation presents an opportunity to develop novel carriers that readily transport therapeutic compounds to the brain.

The outcomes of medication are dependent on the route of administration. It is optimal to choose the administration based on its physiochemical features and interactions with the intended environment, such as enzymes, mucous, and pH [5]. The choice also depends on the amount of drug to be supplied, its solubility, the pH, and the target location [6]. To transport medicine to the brain, intranasal (IN) administration has numerous advantages that overcome the majority of drawbacks associated with parenteral, oral, transmucosal, and direct injections. IN administration is a noninvasive route that absorbs drugs from the nasal cavity via the trigeminal and olfactory pathways and delivers it directly to the brain parenchyma (Figure 1) [7]. For IN delivery, head posture, prior surgical interventions, delivery techniques, and volume affect the deposition of the drug in the nasal passages and its residence time [8]. Potential drawbacks are that some medications may induce nasal discomfort, and there can be poor passage of large-molecule drugs [9]. IN has numerous advantages, including self-administration in humans, ease of administration, noninvasiveness, BBB passage, and a relatively quick onset of effects for fast acting drugs. Recent research has demonstrated that peptide medicines and nanoparticles can be absorbed via nasal delivery [10,11] and thus present an opportunity to develop a range of effective carriers for drugs.

Previous studies have demonstrated that halloysite clay nanotubes effectively penetrate cell membranes due to their small cross-section [12]. In this case, halloysite penetrability is defined by their tens of nanometers diameter rather than their sub-micrometer length (Figure 1). These nanotubes are formed by 10–20 revolutions of 0.72 nm thick aluminosilicate sheets and have diameters ranging between 40 and 60 nm, lumen diameters of 10–20 nm, and lengths within 300–900 nm [13,14,15]. This nanomaterial is inexpensive and available in large quantities. Their outer surface is composed of SiO_2_, and the tube’s interior is composed of Al_2_O_3_, which are negatively and positively charged, respectively, within the pH range of 3–9. Their tube-like structure (Figure 2), the size of the lumen, and their distinct regions of electrical charge, which can be manipulated, make them well suited for loading and delivering a wide variety of drugs.

Based on the dimensions of halloysite, the maximal volume load inside the tubes would be about 10–12 vol.%, which may reach 15–20 wt.% with external drug adsorption. Drugs such as oxytetracycline, vancomycin, dexamethasone, doxorubicin, and furosemide, as well as therapeutic DNA and viral genes, have been successfully loaded into halloysite nanotubes [16].

Halloysite is a biocompatible material with low toxicity, as shown in several studies [17,18,19,20,21,22,23,24]. Their consensus is that these clay nanotubes are safe up to 0.5 mg/mL in tissue, which is less toxic than a common table salt [17]. Halloysite has been tested in both in vitro and in vivo systems, such as cell lines, nematodes, infusoria, fish, mice, and rats [16]. An MTT assay was performed on endothelial cell lines as well to assess their cellular metabolic rate after the addition of halloysite. This was not affected at concentrations up to 50 µg/mL [25]. The only minor toxic effect was found with high oral halloysite consumption due to acidic clay decomposition in the stomach, which increased Al^3+^ accumulation [21]. Mice that were orally fed low nano clay doses (5 mg/kg of mouse mass) exhibited no oxidative stress or other signs of toxicity and even demonstrated higher growth rates. This dose corresponds to only 3 mg of halloysite consumption/day for one month in an adult human.

In this study, we hypothesized that halloysite would pass through the BBB to deliver drugs when administered intranasally to mice. We used dye labeled halloysite to determine that they permeated the BBB and remained in the brain for over 48 h. We also administered halloysite loaded with one of two sedative drugs, diazepam or xylazine, for six days and observed expected deficits in behavioral tests versus vehicle-treated (sham) mice. Behavioral tests were conducted 3.5 h after administration when these drugs would normally have greatly reduced levels when delivered alone. As expected, mice treated with a drug alone had no deficits or less of a deficit versus drug treatment delivered via halloysite. This strongly suggested that the halloysite loaded with a drug permeated into the brain and acted as a slow-release agent. Another important observation was that IN delivery of low doses of halloysite alone did not result in any impairments in behavior. This dose was lower than the consensus level established for safety in other models.

## 2. Results

### 2.1. Quantification of Dye and Drugs Labeled/Loaded into Halloysite

Thermogravimetric analysis (TGA) of major phase transitions was utilized to determine the existence of encapsulated compounds or those absorbed to the surface of the halloysite tubes. In this study, we labeled halloysite with rhodamine B isothiocyanate (RITC) dye for visualization in the brain, diazepam, or xylazine. Diazepam and xylazine are drugs that have sedative effects in mice. The phase transition of halloysite by itself was 500–550 °C, while pristine halloysite nanotubes loaded or labeled with cargo (RITC dye, diazepam, or xylazine) underwent major phase transitions between 200 and 400 °C (Figure 3). The estimated RITC labeling was calculated at 10 ± 2 wt% and 8 ± 1 wt% for diazepam. However, xylazine had a value of 15 ± 2 wt%, which could be due to some external drug attachment to the surface of the nanotube.

### 2.2. Visualization of Halloysite Delivered Non-Invasively to the Brain

The concentration of halloysite and the route of non-invasive administration was determined using halloysite nanotubes labeled with RITC dye in water and emitted light at a wavelength of 550–570 nm. The tubes were administered via IN and intraperitoneal (IP) methods to mice at various concentrations beginning at 0.5 mg/mL. Even at low concentrations of 1 mg/mL, fluorescence was observed throughout the brain in mice that were sacrificed four hours after treatment. A 3 mg/mL dose delivered via IN administration had suitably high levels of fluorescence (Figure 4) versus IP delivery (Appendix A). Consequently, this concentration and route of delivery were utilized in the study. In contrast, no fluorescence was observed for halloysite administration, and very low levels of fluorescence were observed after treatment with the dye alone.

To determine how long halloysite labeled with RITC could be observed in the brain after IN administration, additional mice were treated and sacrificed at 12, 24, 48, and 72 h after a single 3 mg/mL dose (Appendix A). Rhodamine fluorescence remained high through 12 h and then began to decrease through 48 h after treatment. The fluorescence intensity of RITC staining in the olfactory bulb was comparatively higher than in the cortex in mice imaged at 4 and 12 h post-treatment. However, fluorescence was more consistent across the brain at the 24 and 48 h time points. No fluorescence was observed at 72 h (Appendix A) and 7 days post-treatment. Mice given unloaded halloysite showed no fluorescence at any time point, and mice treated with RITC only exhibited relatively weak fluorescence during the first 12 h compared with mice treated with HNTs/RITC.

### 2.3. Performance in Rotarod Test Demonstrates Efficacy of Halloysite Drug Delivery

Both diazepam and xylazine were expected to cause deficits in vestibulomotor performance in a rotarod test, in which a mouse was placed on a rotating cylinder that accelerated. Normal mice were expected to remain on the rotating rod for several seconds before falling off (latency), and impaired mice were expected to have much shorter latency times. To reduce inter-animal variability (and thus the number of mice needed for the test), the scores from days 2, 5, and 7 were normalized with those from the baseline performance prior to treatment.

Mice performed behavioral tests 3.5 h after treatment to allow sufficient time for the clearance of most of the diazepam in the brain for mice treated with diazepam only. This controlled for the possibility that diazepam could be released from the halloysite nanotubes in the nasal cavity, where it could pass into the brain instead of halloysite loaded with diazepam entering the brain. However, the HNT/diazepam-treated mice had significantly poorer performance than the diazepam-treated mice (Figure 5). This supported the hypothesis that halloysite acted as carriers to cross the BBB and deliver diazepam over time. A non-significant decrease in the mean performance of the diazepam-only-treated mice versus the sham and non-loaded halloysite (pristine HNT) groups indicated that some diazepam remained in the brain at the time of the tests, but this group’s performance was still significantly better than the group receiving halloysite loaded with diazepam (HNT/diazepam). In addition, the HNT/diazepam group performed significantly worse than the sham and the pristine HNT-only groups. Notably, there was no significant difference in performance between the sham and pristine HNT-only groups.

The mice treated with xylazine or xylazine loaded into halloysite nanotubes (HNT/xylazine) were not trained prior to treatment to determine if these mice had impaired learning (Figure 6). The mice treated with HNT/xylazine could not learn the task over 7 days. In contrast, the mice treated with xylazine only had a markedly higher latency to fall. Thus, the delivery of xylazine using halloysite tubes appeared to maintain levels of the drug that were high enough to impair learning in this task over the 7-day period. Tests were conducted 3.5 h after treatment using the same rationale as the mice treated with diazepam. Thus, it was expected that the mice treated with xylazine only would have few or no deficits because the level of the drug was relatively low by the time the test was conducted.

### 2.4. Modified Neurological Severity Scores Demonstrate the Efficacy of Halloysite Drug Delivery to CNS

A neurological severity assessment was performed using the set of tests for the modified neurological severity score (mNSS), which is often used to assess the severity of a brain injury in mice [26]. This test was conducted along with the rotarod test on the same days. The purpose of this test was to identify any neurological deficits in mice following treatment. Mice with no impairment generally have a score of zero. The data revealed that the mean scores of the groups of mice treated with pristine HNTs or diazepam alone did not vary from the sham group (Figure 7). In contrast, the group treated with HNT/diazepam had a higher mean score than the other three groups, which suggested that halloysite tubes delivered diazepam to the brain. Most of the mice that failed the test were unable to balance on one or more of the thin rods due to diazepam-induced drowsiness. This was also observed in mice treated with HNT/xylazine (Figure 8).

### 2.5. Anxiety Reduced by HNT/Diazepam Administration

Diazepam is an anxiolytic drug; thus, it decreases the amount of anxiety when a mouse is on an elevated maze. A key indicator of the relative anxiety level is the percentage of time spent in a stretched position to assess the environment. This is commonly referred to as stretch attend posture (SAP). Compared with the sham group, the diazepam group, and the pristine HNT-only-treated group, the mice treated with HNT/diazepam exhibited a decreased percentage of SAP during an elevated maze test (Figure 9). To a lesser extent, the mice treated with diazepam alone had a significantly lower percentage of SAP versus the sham and HNT-only groups (*p* < 0.05). Although the experiments were run at a time when most of the directly administered diazepam would have been cleared from the brain, the repeated administration of diazepam caused its elevation for 24 h or longer [27]. Despite this moderate effect, the HNT/diazepam treatment resulted in significantly lower SAP%, which suggested that the halloysite delivery exerted a greater and more sustained delivery of the drug over time.

### 2.6. Treatment Did Not Alter the Mass of Mice

The mice were weighed before treatment and before they were euthanized. There was no difference in mass between the treatment groups before euthanasia.

## 3. Discussion

Halloysite nanotubes are emerging as potential drug-delivery molecules due to their wide availability, sustained drug-release capabilities, and ability to transport a variety of molecules [29]. These clay nanotube surfaces have been modified, and their effectiveness in targeted therapy has been demonstrated by others [30,31]. Additionally, HNTs have been used to deliver a topical application of drugs to treat infected burns. This resulted in marked improvement in treating the infection and wound healing compared with traditional treatments [32]. Despite testing across a broad spectrum of drug delivery applications, they have not been evaluated for their ability to pass through the BBB until this study. In addition, there is a paucity of research on halloysite in terms of potential adverse effects associated with their administration to the brain.

We examined the ability of halloysite nanotubes to penetrate the BBB and release drugs in the brain and whether halloysite alone caused any changes in behavioral tests. In this study, IN infusion was more effective at delivering rhodamine-labeled halloysite to the brain than intravenous injection. Halloysite appeared to remain in the brain for about two days. After 48 h, there was little fluorescence in the RITC/labeled samples, and by the seventh day, no fluorescence was observed. The lack of fluorescence observed could be attributed to the elimination of halloysite and RITC from the brain, or the RITC discharged by the nanotubes may have been eliminated by the inherent cleansing mechanism of the cerebral spinal fluid.

Our results demonstrated that the administration of halloysite nanotubes without drugs did not result in any behavioral deficits in locomotor learning and anxiety tests. When the halloysite nanotubes were administered intranasally, a few animals exhibited mild discomfort (gritting teeth) for a couple of minutes. This could have been the result of a temporary nasal irritation.

Diazepam is an anxiolytic drug, and upon the administration of higher dosages, it will cause sedation and impair locomotor abilities. The sedative, xylazine, also would also result in poorer short-term performance in locomotor tests (e.g., mNSS and rotarod). Each would have less of an effect on performance after about three hours [33], which is why behavioral tests were conducted at 3.5 h after administration. At this time point, there would be a significant difference if the HNT/drugs passed through the BBB and remained there to release the drugs over time. The significant differences observed after the administration of diazepam or xylazine loaded with HNTs revealed that not only did they permeate the BBB, but they released the drugs over time in the brain. This was consistent with previous studies of halloysite-assisted drug delivery in other types of tissue [14,26].

In conclusion, halloysite nanotubes have the potential to carry drugs across the BBB using clinically relevant, noninvasive IN administration and to release the molecules over time in the brain. Further studies in mice must be conducted to determine the fate of halloysite nanotubes once they enter the brain, the effects of their long-term administration, and if a maximum dose level is indicated. If future studies reveal the halloysite nanotubes are not completely eliminated or are eliminated slowly, they might still be utilized in humans for chemotherapy or for acute, emergency treatment for conditions such as traumatic brain injury or stroke where prompt treatment can prevent secondary injury [34].

## 4. Materials and Methods

### 4.1. Chemicals

Halloysite was purchased from Sigma-Aldrich (St. Louis, MO, USA) and used without further purification. Rhodamine B isothiocyanate (RITC), 4% formalin, and dibasic sodium phosphate were also obtained from Sigma-Aldrich. Ketamine hydrochloride (Vedco, Saint Joseph, MO, USA), 0.9% saline (Teknova, Hollister, CA, USA), and xylazine (Vetone) were used to prepare a 10% k/x solution. The other used chemicals and solutions were phosphate-buffered saline (PBS, Gibco, Billings, MT, USA), 37% formaldehyde (Ward’s Science, Rochester, NY, USA), monobasic sodium phosphate (MP biomedicals, Santa Ana, CA, USA), 4′,6-diamidino-2-phenylindole, dilactate (DAPI, Roche Diagnostics, Basel, Switzerland), Glycerol (Himedia, Maharashtra, India), and n-propyl gallate (MP Biomedicals).

### 4.2. Instrumentation

The experimental analyses were conducted with the following systems and scanning and transmission electron microscopes (EDAX-SEM, Hitachi-S4800, Tokyo, Japan and TEM, JEM-2100, JEOL, Tokyo, Japan, Hitachi HD 2000 STEM, Claremont, CA, USA) as well as thermogravimetric analysis (TGA, Thermal Advantage Q50, Newtown, PA, USA), a UV-vis spectrophotometer (Agilent 8453, Santa Clara, CA, USA), and fluorescent and laser confocal optical microscopy (Leica DMI 6000 B inverted microscope, Deerfield, IL, USA and Nikon A1R Confocal Super Resolution System, Melville, NY, USA).

### 4.3. Preparation of Halloysite with Labeling/Loading of RITC, Diazepam, and Xylazine

These formulations were conducted using halloysite from Applied Minerals. A ratio of 1:2 halloysite to RITC was prepared. Once the initial measurements were concluded, the samples were mixed in DI water through ultra-sonication for 10 min. The sonication allowed the halloysite to disperse properly, initiating the process of creating a super-saturated solution for maximum dye encapsulation. Notably, no external surface modification (i.e., polymer coating) was used in this or our other formulations. After sonication, the solution was stirred on a magnetic stir plate for 24 h at 20–22 °C. Washing and centrifugation were used to remove any excess dye that could be attached to the outer shell of the tubule. The centrifugation included three repetitions at 4000 rpm for 3 min. After this, the sample was dried at 70 °C for 24 h.

A 2:1 ratio of diazepam with pure, unmodified halloysite was utilized. Initially, the diazepam tablets were milled into a fine powder and then combined with pristine halloysite during ultrasonication in DI water to induce halloysite dispersibility. A super-saturated solution was then formed through stirring for 24 h at 20–22 °C. The solution was then centrifuged three times at 4000 rpm for 3 min. This process removed excess diazepam that could have been adsorbed to the outer shell of the nanotubes. Afterward, a desiccator vacuum was used to remove any remaining moisture from the sample over 24 h. The final product was then milled and used for thermogravimetric analysis (TGA), which is described below.

Utilizing a similar strategy, a 2:1 ratio of pure unmodified halloysite was sonicated with the introduction of xylazine (100 mg/mL) for 10 min to induce the dispersibility of the halloysite and induce the effective uptake of the drug. The mixture was then stirred for 24 h at 20–22 °C, creating a super-saturated solution. Centrifugation was then performed to remove any excess xylazine that could be attached externally to the halloysite. This process involved three rotations at 4000 rpm for 3 min. A desiccator vacuum was then used to remove any remaining moisture from the sample over 24 h. The final product was then milled and used for thermogravimetric analysis.

TGA (Thermal Advantage Q50, USA) was used to determine the change in the mass of the loaded HNT during component burning. The loss in mass represented the drugs (or dye) that was loaded in the halloysite and the approximate loading percentage. A 5 mg desiccated sample of the loaded HNT was placed in an aluminum pan, and then the contents of the HNT were burned up to a maximum of 600 °C for a total of 1 h. During the burning process, multiple peaks were displayed at each temperature where material had been burned. The difference in weight change % from each sample at respective temperature points was calculated using the proprietary software (Universal V4.5A) supplied with the TGA system.

### 4.4. Animal Care and Handling

Animal handling and care and the experimental procedures were conducted according to a protocol approved by the Louisiana Tech University Institutional Animal Care and Use Committee. Wild-type C57BL/6NHsd were purchased from Jackson Laboratory and bred according to the need. Animals were maintained with 12 h dark/light cycles and provided with food and water ad libitum.

### 4.5. Intranasal Administration

The mice were separated into individual cages throughout treatment. The mice were sedated in an anesthetic chamber with 2% isoflurane in 500 mL/min airflow using a SomnoSuite digital vaporizer system (Kent Scientific Corporation, Torrington, CT, USA). During the intranasal delivery of treatment solutions, light anesthesia was maintained using a nose cone with 1% isoflurane. An IN infusion of 6 µL of treatment solution was placed by micropipette into one nostril while in a supine position at 70 degrees, followed by three more treatments administered into alternate nostrils with one minute between each treatment. The mice were weighed before the first dose and before sacrificing to determine if the treatment affected weight gain or loss. Three female mice and three male mice (between 8 and 16 weeks) were randomly assigned to each treatment group. To evaluate the BBB permeability, we administered 2 mg/mL halloysite labeled with RITC dye, 1 mg/mL RITC dye alone, or 1 mg/mL pristine halloysite alone. To evaluate the delivery of drugs across the BBB using halloysite, we used the following treatment groups: (1) sham treatment with vehicle (water), (2) HNTs loaded with 3 mg/mL of diazepam or xylazine, (3) 2 mg/mL of diazepam or xylazine without HNTs, or (4) 2 mg/mL of pristine HNTs only. The treatments were administered once daily at the same time of day for six days. After receiving the treatment, the mice were returned to their home cage and observed for at least 10 min for any abnormal behavior.

### 4.6. Intraperitoneal Administration

To determine the permeability of HNTs to the brain using intraperitoneal administration, mice were given 10% *w/v* of 2 mg/mL HNT and sacrificed 4 h later. Once the mouse was completely sedated, it was positioned supine and given an intraperitoneal dose before being returned to its cage [35]. The other procedures for handling mice were the same as described for IN delivery.

### 4.7. Evaluation of BBB Permeability

Mice were sacrificed at 4, 12, 24, and 48 h and 3 and 7 days after a single treatment with RITC/HNT (2 mg/mL) to assess the permeability of halloysite across the BBB. The mouse was briefly anesthetized with 2% isoflurane at 500 mL/min using a SomnoSuite digital vaporizer to reduce stress and pain from a subsequent injection of an anesthetic cocktail. Once anesthetized, an anesthetic cocktail of 10% *w/v* of ketamine (10 mg/kg)/xylazine (1 mg/kg in saline) was administered intraperitoneally [34,35,36]. The mouse was returned to its home cage to reduce stress before the anesthetic cocktail exerted its effect.

Once the mouse was deeply anesthetized, its forelimbs and tail were taped to the surgery table while it was laid in a supine position to expose its abdomen. An incision was made to expose the thoracic cavity, and then a small incision was made in the right aorta to facilitate blood drainage. Then the mouse was perfused with 25 mL of 1X ice-cold PBS and then 25 mL of 4% ice-cold formalin utilizing a peristaltic perfusion pump with a needle inserted into the left ventricle. Once the liver was cleared, the mouse was decapitated, and the brain was removed [36]. The brain was then post-fixed in 4% formalin for eight hours at 4 °C. After this, it was transferred to a 30% sucrose solution until the tissue floated, and then it was frozen and stored at −80 °C.

Brains were sectioned using an EMS-500 oscillating tissue slicer (Electron Microscopy Sciences, Hatfield, PA, USA). After removal of the brain stem to create a flat base, the brain was encased in a 2% solution of low-melting agarose. After the removal of the excess agarose on the cut side, the tissue was adhered to the mounting block using cyanoacrylate glue, and 60 µm sections were cut. Throughout the process of slicing, the brain was submerged in an ice-cold buffer solution. The slices were transferred with care to a 48-well plate containing PBS solution and stored in the dark at 4 °C until the preparation of slides.

The tissue sections were stained for nuclei visualization with DAPI for 10 min at 4 °C on a shaker, followed by optical clearing with varying glycerol concentrations. The sections were soaked for 10 min in 30%, 50%, 70%, and 90% concentrations of glycerol. After saturating the sections, they were mounted on slides containing anti-fade mounting media. The sections were arranged from rostral to caudal order, including the olfactory bulb and different regions of the cortex. After mounting, the slides were coverslipped, sealed with nail polish, and stored in the dark.

An Olympus IX51 epifluorescent microscope was used to acquire the phase contrast, DAPI, and RITC images of brain sections while maintaining the same settings for all images. Randomly selected regions of the same section were imaged, and several measures were taken to prevent bleaching of the section during imaging, including optimizing the light intensity and imaging time and the acquisition of RITC images, followed by DAPI and then phase contrast images for each region of interest.

### 4.8. Rotarod Test

A Panlab Model 76-0770 rotarod system (Harvard Apparatus, Holliston, MA, USA) was used for all tests. This test assessed the locomotor function, coordination, and learning capacities of mice. The rotarod was thoroughly cleaned between each mouse. Prior to each test, mice were acclimated to the environment in the testing room for at least 10 min to reduce stress. Each test day, three consecutive trials for each mouse were conducted with a 10 min rest between the second and third trials. The training for the rotarod test began three days prior to the administration of the treatments and was conducted daily (days -3, -2, and -1) at the same time of day. The mean time to fall (latency) of the two best tests for each mouse was averaged and recorded on the third day of training (day -1) as the animal’s baseline score. The performance was then evaluated on tests conducted 2, 5, and 7 days after the initial administration of treatments (days 2, 5, and 7). For each test day, the best two of three scores for each mouse for each day were averaged, and then these mean scores were normalized to the animal’s baseline score [34,37].

### 4.9. Modified Neurological Severity Score (mNSS)

The mice were assessed on days 2, 5, and 7 after treatment began. The following eight behaviors were evaluated: hindlimb flexion, startled response, seeking behavior, the capacity of mice to walk on elevated bars 3 cm, 2 cm, and 1 cm broad and 30 cm long, and the ability to balance on a 0.5 cm width square beam (at least for 30 s) and a 0.5 cm thick circular rod. mNSS was used to assess the aberrant impaired balance, reflexes, and visual, proprioceptive, and tactile sensory functions [35,38]. Each failure in a task resulted in one point being added to the mNSS. A score of zero was expected for unimpaired mice, whereas a score of eight indicated severe neurological dysfunction.

### 4.10. Stress Attend Posture in the Elevated Plus Maze

The tests were conducted in a temperature-conditioned environment on a 50 cm high maze with two open arms and two closed arms on day 7 from the start of treatments. These arms were positioned at right angles to one another, and their closed walls measured 25 cm in height. Prior to the testing, the mice were acclimated to the room, and the maze was cleansed to remove any odors from the mice that were previously tested. Each mouse was placed in the center of the maze with its back to the open arms, and a video recording was made as the mouse explored the maze for 5 min. Recordings were made from an overhead video acquisition system.

The video recordings were used to analyze a natural behavior associated with risk assessment called stretch attend posture (SAP), in which a rodent’s rear feet remained stationary with its body stretched forward; this position was maintained for a brief moment. The video files were converted to audio video interleave (AVI) uncompressed files at 10 frames per second. The final frame, without the mouse, was used to set the intensity of the background for the subtraction of noise. The video files were edited to precisely 3000 frames, and the audio channel was removed using Image J software [29]. The necessary adjustments, such as altering the angles to fit the square, were made, and then these files were converted to multi-TIFF files, which were then read by MATLAB plugins. We then used MATSAP, a MATLAB-based program, to identify and analyze this behavior that was previously developed in our lab [29,35].

### 4.11. Statistical Analysis

The statistical analysis was conducted using SPPS software (version 28.0.1.0). ANOVA with a Bonferroni correction for multiple comparisons was used for the experiments with four treatment conditions with α set at 0.05. Student *t* tests were used for pairwise comparisons, and *p* < 0.05 was set as the level for significance.

## Data Availability

Supporting results are available at https://doi.org/10.6084/m9.figshare.23254601.

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
