# Peer review of "Clay Nanotubes Loaded with Diazepam or Xylazine Permeate the Brain through Intranasal Administration in Mice"

_ijms, 2023, doi:10.3390/ijms24119648_

Round 1

Reviewer 1 Report

In the present manuscript the authors investigate the intranasal administration in mice, of halloysite loaded with different drugs for brain treatment. The topic is very interesting and progressive and it represents a step forward to the use of halloysite as carrier systems, considering the fact that the clay cannot be administered by intravenous injection. In my opinion the manuscript can be published on International Journal of Molecular Science after minor revision. See below for further details:

1. Were there any impurities in these halloysite clay nanotubes, like kaolinite? Would you be able to provide an additional SEM image to display the quality of these tubes (better than Fig. 1 C)?

2. Please label the trigeminal and olfactory nerves in Scheme 1.

3. Figure 2. Does the arrow point to the lumen?

4. For the RITC dye, what is the wavelength of emitted light?

5. The phrase “and then by its exit” is not clear (Line 285). Consider revising the phrase. The word “administration” is misspelled in Line 295. The words in “[add citation 36]” on Line 409 should be deleted. For Line 420, what does DAPI stain? Furthermore, the term halloysites should be changed with halloysite through the manuscript, as well as the terms referred to halloysite as tubules should be changed with tubes.

Finally, text should be uniformed, for example line 141 and so on the authors should refer to RTC as labelling molecule for HNT, therefore they should use the term labelled instead of loaded or complexed.

Line 158 change HNT-RTC in HNT/RTC. Similar considerations for all drugs considered in the study, since they are supramolecular loaded, I suggest the authors to use slash instead of hyphen.

6. Line 187, what the authors refer to HNT-only groups? Please specify. Did they mean pristine HNT? It the answer is yes, I suggest them to use the term pristine HNT.

7. Some references about the use of halloysite as drug carrier should be added in the introduction part to better highlight the importance to use the clay in biological field. See for example: J. Coll. Interf. Sci. 2022, 620, 221-233, Coll. Surf. B, 2022, 213, 112385, Int. J. Pharm. 2021, 599, 120281.

The english is fine

Reviewer 2 Report

The study is of interest for the ijms readers but the manuscript needs to be revised, especially the figures for the statistical issues. The supplementary figures were not accessible, the reviewer cannot give their opinion.

1) To the reviewer’s opinion, Figures 5, 7 and 9 should be revised for the statistical significance of the results. The significance for a p value of 0.001 is better that 0.01 and better than 0.05. The reviewer suggest to use for those figures the following scale:

* p<0.05     ** p<0.01   *** p<0.001    **** p<0.0001

The legends to those figures must be adjusted to this use. As an example : for figure 7 you could indicate * for all the comparisons and in the legend (* p<0.05, n=6 mice per group)

2) Figure 6: The indication ‘*’ makes no sense, they should be removed. The information about the p value in the legend is enough. Also, in the legend, the following sentence makes no sense “The time on the rod before failing (latency) to reduce the effect of inter-animal variability”.

3) It will be nice in the discussion to explain what infers to mice, what infers to humans or at least make a projection on what one can expect for the use of halloysites for humans, if any study give information about their toxicity.

Others points:

line 44: indicate 'parenteral' instead of 'parental'

lines 45 and 142: indicate 'IN' instead of 'Intranasal', the abbreviation was already defined line 43

Scheme 1: 'Pathway' instead of 'Pathways'. Also, in the caption the yellow arrows on a yellow background are not visible

Round 2

Reviewer 2 Report

The reviewer thank the authors who addressed all their recommendations. The manuscript is now acceptable for publications.